# *Enterococcus faecium* Modulates the Gut Microbiota of Broilers and Enhances Phosphorus Absorption and Utilization

**DOI:** 10.3390/ani10071232

**Published:** 2020-07-20

**Authors:** Weiwei Wang, Huiyi Cai, Anrong Zhang, Zhimin Chen, Wenhuan Chang, Guohua Liu, Xuejuan Deng, Wayne L. Bryden, Aijuan Zheng

**Affiliations:** 1Key Laboratory for Feed Biotechnology of the Ministry of Agriculture and Rural Affairs, Institute of Feed Research, Chinese Academy of Agriculture Sciences, Beijing 100081, China; dbnywzw@163.com (W.W.); caihuiyi@caas.cn (H.C.); 15504578605@163.com (A.Z.); chenzhimin@caas.cn (Z.C.); changwenhuan@caas.cn (W.C.); liuguohua@caas.cn (G.L.); 2National Engineering Research Center of Biological Feed Development, Beijing 100081, China; xjdeng2004@126.com; 3School of Agriculture and Food Sciences, University of Queensland, Gatton, QLD 4343, Australia; w.bryden@uq.edu.au

**Keywords:** broiler, phosphorus, *Enterococcus faecium*, microbiota, 16S rDNA

## Abstract

**Simple Summary:**

Bone health is an important factor in broiler production. Among the key nutrients affecting bone health, phosphorus (P) plays a great role. *Enterococcus faecium* has been widely used as feed additive to promote growth performance of broilers. There were reports suggesting that *E. faecium* improved skeletal health of rats. However, the effect of *E. faecium* on the bones of broilers remains unclear. The present study is to investigate the effect of *E. faecium* on P absorption and utilization in broilers and the associated changes in the gut microbiota. Dietary inclusion with *E. faecium* did not improve broiler performance in this study but improved P absorption and bone mineralization. In *E. faecium*-treated broilers, the expression of intestinal type IIb sodium-dependent phosphate cotransporter (*NaP-IIb*) mRNA was upregulated and the concentration of serum alkaline phosphatase was increased. Dietary supplementation with *E. faecium* changed the gut microbiota populations of broilers and increased the relative abundance of SCFA (short-chain fatty acid)-producing bacteria. The changed populations of microbiota improved intestinal P absorption and bone forming metabolic activities. In conclusion, dietary inclusion with *E. faecium* facilitates increased utilisation of P in broilers.

**Abstract:**

Modern broiler chickens have ongoing bone health problems. Phosphorus (P) plays an important role in bone development and increased understanding of P metabolism should improve the skeletal health of broilers. *Enterococcus faecium* has been widely used as a probiotic in broiler production and is shown to improve skeletal health of rats, but its effect on the bones of broilers remains unclear. This study investigated the effect of *E. faecium* on P absorption and utilization in broilers and the associated changes in the gut microbiota using 16S rDNA sequencing. Dietary supplementation with *E. faecium* improved P absorption through upregulation of the expression of intestinal *NaP-IIb* mRNA and increased the concentration of serum alkaline phosphatase. These actions increased P retention and bone mineralization in *E. faecium*-treated broilers. The positive effects of *E. faecium* on P metabolism were associated with changes in the populations of the intestinal microbiota. There was increased relative abundance of the following genera, *Alistipes*, *Eubacterium*, *Rikenella* and *Ruminococcaceae* and a decrease in the relative abundance of *Faecalibacterium* and *Escherichia-Shigella*. Dietary supplementation with *E. faecium* changed gut microbiota populations of broilers, increased the relative abundance of SCFA (short-chain fatty acid)-producing bacteria, improved intestinal P absorption and bone forming metabolic activities, and decreased P excretion. *E. faecium* facilitates increased utilisation of P in broilers.

## 1. Introduction 

Skeletal disorders and associated welfare problems are an ongoing issue for fast growing broiler chickens and a major concern throughout the global poultry industry [1,2]. Calcium (Ca) and phosphorus (P) are the most important minerals in bone development and comprise the inorganic component of bone tissue, providing hardness and strength to the skeleton [3]. Diets deficient or imbalanced in these co-dependent minerals severely decrease growth performance and nutrient retention of broilers [4,5]. Many studies have investigated the absorption and metabolism of Ca and P, and there is a greater understanding of the regulatory mechanisms controlling Ca metabolism than P metabolism [5]. In addition, eutrophication due to high P excretion is becoming more and more serious, which intensifies concerns about P utilization and the sustainability of broiler production [6].

The addition of probiotics to poultry diets has increased significantly in recent years as the use of antibiotic growth promoters has declined [7]. This has prompted much research into the use of new probiotic feed additives. *E. faecium,* a lactic acid bacterium and normal inhabitant in the gut, is a probiotic that can promote growth performance, can reduce mortality, can improve intestinal morphology and can beneficially modulate the gut microbiota of broilers [8,9,10]. These characteristics of *E. faecium,* along with the ability to increase the efficiency of intermediary metabolism [11] and to improve meat quality [12], have made it an attractive poultry feed additive. Moreover, some probiotics have shown beneficial effects on the skeletal health of broilers [13,14] and rodents [15]. These observations are consistent with probiotics modifying the gut environment, including the gut microbiota, and/or enhancing mineral absorption. Although the mode of action(s) of probiotics are poorly understood [7], modulation of the gut microbiota is likely to be important.

There is little research on the effect of *E. faecium* on bone health. However, it has been suggested that *E. faecium* can prevent whole body bone mineral density loss in arthritic rats [16]. It is therefore likely that *E. faecium* will have some positive effect on the bone health of broilers. The objective of the present study was to investigate the effects of dietary *E. faecium* on performance traits, bone strength, P absorption and gut microbiota of broilers and to explore the regulatory mechanism of *E. faecium* on P absorption and utilisation in broilers.

## 2. Materials and Methods

### 2.1. Ethics Statement

Feeding trials were conducted according to the guidelines for animal experiments set out by the National Institute of Animal Health, and all animal procedures were approved by the Chinese Academy of Agricultural Sciences (statement no. AEC-CAAS-20191106).

### 2.2. Experimental Design

A total of 120 1-day-old male, Arbor Acres (AA) broilers, were purchased from the Huadu Broiler Breeding Co. (Beijing, China) and housed in the Nankou experimental farm of the Feed Research Institute, CAAS, Beijing, China. The day-old chicks (body weight, 47.2 ± 0.31 g) were randomly divided into two groups: control and treatment. Each group had 6 cages (replicates) with 10 birds per cage. The chickens were reared in two stages, starter (1–21 days) and grower (22–42 days), and fed a basal (control) corn-soybean meal diet (Table 1) in pellet form, to which 6.75 × 10^9^ cfu/g of *E. faecium* was added before pelleting for the treatment group. Microcapsules of *E. faecium* CGMCC 2516 (viable count ≥15 × 10^10^ cfu/g; Challenge Group, Beijing, China) were used in this study. *E. faecium* CGMCC 2516 were cultured in de Man Rogosa and Sharpe (MRS) medium, adding suitable concentrations of anhydrous calcium chloride. The culture was dried under the condition of 45 °C and was solid microencapsulated using coating technology [17,18].

### 2.3. Bird Management

Birds were raised in accordance with the AA Broiler Management Guide [19]. Chicks were vaccinated for Marek’s disease at day-old and for Newcastle disease and infectious bronchitis at 7 days post-hatching. Room temperature was maintained at 33 °C for days 0–3 and was gradually reduced to 24 °C and maintained at 24 °C till the end of the study. The photoperiod was controlled to 16 h of light and 8 h of darkness. Relative humidity was set at 60–70% during the first week and then at 50–60% for the rest of the experiment.

### 2.4. Sample Collection and Parameter Determination

From days 18 to 21 and days 39 to 42 of the experiment, excreta from each replicate was collected, mixed and dried in an oven at 105 °C to a constant weight. The dried excreta were ashed in a muffle furnace at 550 °C for 4 h. The P content of the ash samples was determined using the vanadate-molybdate method [20].

On day 21 and day 42, body weight (BW) and feed intake were measured to calculate the average daily gain (ADG), average daily feed intake (ADFI) and the ratio of feed/gain (F/G). On those days, one broiler close to the cage average body weight was randomly selected from each replicate. The chosen birds were electrically stunned and manually slaughtered within 5 min [21]. Blood was collected from the jugular vein, and serum was obtained after centrifuging at 3000× *g* for 10 min at 4 °C and stored at −20 °C for further analysis. Serum alkaline phosphatase (ALP) and P were determined with a Hitachi 7600 automatic biochemical analyser using kits purchased from Nanjing Jiancheng Biological Engineering Institute.

After the blood sampling on day 42, the duodenum (about 10 cm distal to the pylorus), jejunum (about 10 cm preceding the Meckel’s diverticulum) and ileum (about 10 cm preceding the ileocecal junction) were separated [22] and flushed gently with saline solution. The mucosa samples were scraped with a coverslip and snap-frozen in liquid nitrogen for analysis of mRNA.

The right tibiae were cleaned and dried for determination of tibia weight and tibia breaking strength [23]. The bones were then ashed in a muffle furnace at 550 °C for 16 h [20]. After that, the tibia P content was measured using the vanadate-molybdate method.

### 2.5. RNA Extraction, Reverse Transcription and Real-Time Quantitative PCR

Total RNA was extracted using TRNzol-A + (TIANGEN, Beijing, China). The concentration of total RNA was estimated by a spectrophotometer (Ultrospec 2100 pro, GE Healthcare, Chicago, IL, USA), and the purity was determined by agarose gel electrophoresis. Five hundred nanagrams of total RNA were reverse transcribed into cDNA using the Fast Quant RT Kit (with gDNase) (TIANGEN). qPCR was conducted using the iCycler iQ5 system. The specific primers for *NaP-IIb,* type III sodium-dependent phosphate cotransporter-1,2 (*PiT-1, 2*) and *β-actin* are listed in Table 2. *β-actin* was used as internal reference gene. Relative gene expression was calculated using the 2^-ΔΔCt^ method [24]. All the samples were analysed in triplicate, and the operational program for qPCR strictly followed the Minimum Information for Publication of Quantitative Real-Time PCR Experiments (MIQE) [25].

### 2.6. Illumina Sequencing Analysis

The faecal samples were collected on day 42 and snap-frozen in liquid N_2_ prior to further processing. Gene sequencing (16S rDNA) was performed by OE Biotech Co., Ltd. (Shanghai, China). Total genomic DNA from frozen faecal samples was isolated using the GenElute™°Stool DNA Isolation Kit (Sigma-Aldrich, St. Louis, MO, USA); then, the V3–V4 hypervariable region of the 16S rDNA genes was amplified. The PCR products were collected and sequenced using the Illumina MiSeq platform (Illumina, San Diego, CA, USA). High-quality reads were clustered into operational taxonomic units (OTUs) based on sequences with ≥97% similarity and then analysed using the QIIME platform.

### 2.7. Statistical Analysis

The statistical analyses were performed using SPSS 17.0. The data were statistically analysed by T-Test (independent samples). For the indexes with significant main effect, Duncan’s method was used to compare the mean values among groups. A *p*-value less than 0.05 was considered significant.

## 3. Results

### 3.1. Growth Performance

The birds grew normally and remained in good health throughout the experiment. The effects of dietary *E. faecium* on growth performance of broilers is shown in Table 3. In the starter stage (1–21 days), dietary supplementation with *E. faecium* decreased (*p* < 0.05) the ADFI of broilers but did not affect (*p* > 0.05) BW, ADG and F/G. In the grower stage (22–42 days), *E. faecium* groups showed a tendency to numerically increase BW and ADG and to decrease ADFI and F/G of broilers.

### 3.2. Ash and P of Excreta, Serum P and ALP Concentrations 

Supplementation with *E. faecium* did not affect excreta the ash content during either growth stage or the P excreta content of starter chicks but did decrease (*p* < 0.05) P excretion in the grower stage (Table 4). No difference in serum P concentration was observed between the two groups of broilers. However, serum ALP increased significantly (*p* < 0.05) in both the starter and grower stages when compared to the control group.

### 3.3. P and Ash of Bone, and Tibia Strength

In the starter stage, P and ash content of bone were not influenced by *E. faecium* supplementation (Table 5), but in the grower stage, the probiotic significantly (*p* < 0.05) increased both parameters, while tibia strength was not affected in the starter or grower stage.

### 3.4. NaP-IIb and PiT-1, 2 mRNA Expressions in the Duodenum, Jejunum and Ileum of Broilers

Dietary supplementation of *E. faecium* significantly increased (*p* < 0.05) *NaP-IIb* mRNA expression levels in the duodenum, jejunum and ileum (Table 6). However, *PiT-1* and *PiT-2* mRNA expressions were not affected in the duodenum and jejunum but increased in the ileum of the *E. faecium*-treated group.

### 3.5. Gut Microbiota Analysis

As revealed by principal component analysis (PCA), dietary supplementation with *E. faecium* changed the populations of the gut microbiota of broilers (Figure 1A). In addition, *E. faecium* led to a significant increase in the observed species and Simpson indices with respect to the control values (Figure 1B,C), suggesting that *E. faecium* exerted stronger positive effects on the α diversity of the gut microbiota of broilers.

The gut microbiota composition at phylum and genus levels is shown in Figure 2. The predominant phyla were *Firmicutes*, *Bacteroidetes*, *Proteobacteria*, *Epsilonbacteraeota*, *Actinobacteria* and *Tenericutes*, representing 59.8%, 35.0%, 4.25%, 0.49%, 0.21% and 0.20% of the total sequences, respectively (Figure 2A). There were no significant differences in phylum level between *E. faecium*-treated and control birds. Compared to the control group, a higher abundance of *Bacteroidetes* and a lower abundance of *Proteobacteria* were observed in the *E. faecium*-treated group. The genus level analysis revealed that *E. faecium* mainly increased the relative abundance of *Alistipes*, *Eubacterium*, *Rikenella* and *Ruminococcaceae*, while the relative abundances of *Faecalibacterium* and *Escherichia-Shigella* were decreased (Figure 2B).

We compared the gut microbiota of the two experimental groups using linear discriminant analysis effect size (LEfSe) to identify the specific microbiota linked to *E. faecium* treatment. *Peptoclostridium, Ruminococcaceae*, *Papillibacter* and *Eubacterium* were more abundant in the *E. faecium* groups (Figure 3A,B).

## 4. Discussion

The current study showed that dietary supplementation of *E. faecium* improved the growth performance of broilers but not significantly. These results were consistent with previous studies which suggested that inclusion of a probiotic had no effects or a slight improvement on growth performance of broilers [9,26]. However, in other studies using various probiotic strains, significant improvement in growth performance of broilers has been demonstrated [27,28,29,30]. The different outcomes in broiler growth performance are a response to many factors including the probiotic strain used and the experimental conditions. Chickens raised in less optimal conditions usually have inflammatory immune reactions in the intestinal mucosa and abnormal gut pH, which would exert bad influence on the growth performance of birds and give opportunities to probiotics to get the largest effect [31,32].

Bone metabolism largely mediates P and Ca metabolism, which are closely related but differ in relation to the endocrine control of absorption and renal reabsorption/excretion [33]. Dietary P is absorbed and accumulated in the small intestinal mucosa and then released into the blood gradually [34], where concentrations are maintained by homeostasis [35], as was evident in the current study. Previous studies in rats, humans and chickens have suggested that probiotics could promote intestinal absorption of P [36,37,38,39]. In this study, *E. faecium* increased the P content of bone. This result reflects increased P accretion in bone, where the bone-forming cells, osteoblasts, are responsible for the deposition of the bone matrix [40]. The concentration of ALP in serum, a marker of osteoblast activity, is usually elevated when bone formation rates are increased [41]. In the current study, ALP levels in serum of the *E. faecium* group were significantly higher than the control in both starter and grower stages, resulting in greater retention of P.

The increase in P deposition reflects both increased absorption and reduced excretion. The decreased concentration of P in excreta may reflect both enhanced intestinal absorption and renal reabsorption. Much of the intestinal absorption of P is accomplished by the sodium-dependent transporter *NaP-IIb*, which is primarily expressed in the duodenum of broilers and is the most important P transporter in the small intestine [42]. In our study, mRNA expression levels of *NaP-IIb* were increased in the small intestine of *E. faecium*-treated broilers, indicating that increased P absorption occurred in this study. The elevation of *NaP-IIb* expression levels in the *E. faecium* group may have resulted from increased available P in the intestine. The expression of *NaP-IIb* mRNA increases when the level of dietary P increases and is concentration dependent [22]. Some probiotics produce phytase [43], which would enhance phytate digestibility, releasing P for absorption. Perhaps *E. faecium* produces a phytase or facilitates increased phytase activity by the intestinal microbiota.

In the current study, the populations of ceacal microbiota was changed in the *E. faecium* group compared with the control group. *E. faecium* treatment resulted in a dramatic elevation in observed species and Simpson indices, indicating that the richness and diversity of the microbiota was significantly increased. Previous studies have shown that inclusion of *E. faecium* in broiler diets beneficially alters the gut microbiota [8,44]. Our results demonstrate that dietary *E. faecium* can modulate the intestinal flora of broilers, as indicated by 16S rDNA-based analysis. As a probiotic, *E. faecium* can exert antagonistic functions via competition for nutrients, metabolites and an occupying effect. In this study, the proliferation of potentially pathogenic bacteria, *Faecalibacterium* and *Escherichia-Shigella*, was inhibited. This is consistent with previous studies in which reduced relative abundance of *Faecalibacterium* and *Escherichia-Shigella* was also observed in *E. faecium*-feeding piglets [45,46]. *Alistipes*, initially isolated from human gut, is a strict anaerobe that produces succinic acid as the principal metabolic end-product of glucose fermentation [47,48]. Mice studies suggest that members of this genus affect host physiology, including sites distal to the gastrointestinal tract [49]. *Rikenella* has been isolated from faeces and caeca of a variety of animals including chickens. It is an obligate anaerobe and yields propionic and succinic acids together with moderate amounts of acetic acid from glucose fermentation [50]. The other two upregulated genera, *Eubacterium* and *Ruminococcaceae*, are generally associated with increased butyrate production [51], which is an important energy substrate for intestinal enterocytes. In general, the upregulated genera in the *E. faecium* group all are associated with increased SCFA production. SCFA production followed by a decrease in gut pH leads to increased mineral solubilization [52], which would help P absorption via the elevation of *NaP-IIb* expression levels because when the content of available P in the intestine is increased, the expression of *NaP-IIb* mRNA will increase accordingly [22]. Bone mineralization enhanced by changes in gut microbiota has also been proven in rats. Report showed that dietary supplementation with synbiotics exerted a synergistic effect on bone mineralization of rats, which was associated with higher counts of SCFA production genera and a reduced pH in the intestine [53]. Additionally, increased circulating concentrations of SCFA can interact with the skeleton to directly inhibit bone resorptive osteoclast differentiation and to activate bone forming osteoblasts, thus increasing bone mass and preventing bone loss [54,55].

## 5. Conclusions

In conclusion, regardless of the mechanism(s), *E. faecium* facilitates increased utilisation of P in broilers. Dietary supplementation with *E. faecium* increased the relative abundance of SCFA-producing bacteria, improved intestinal absorption of P and bone forming metabolic activities, and decreased P excretion. Further research is required to more clearly define the metabolic actions of probiotics to permit their strategic use. The results of this study are, however, another illustration of the benefits of supplementing poultry diets with probiotics.

## Figures and Tables

**Figure 1 animals-10-01232-f001:**
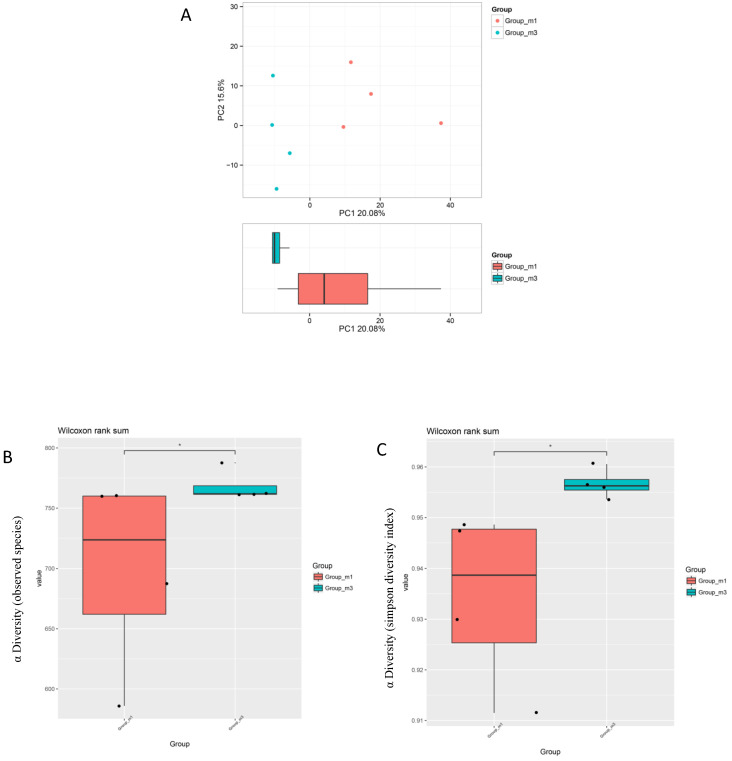
*E. faecium*-modulated gut microbiota of broilers: (**A**) principal component analysis (PCA) scores indicated the difference in gut microbiota populations and (**B**,**C**) observed species and Simpson diversity indexes of the gut microbiota. m1, control. m3, *E. faecium*. ns, not significant. * *p* < 0.05.

**Figure 2 animals-10-01232-f002:**
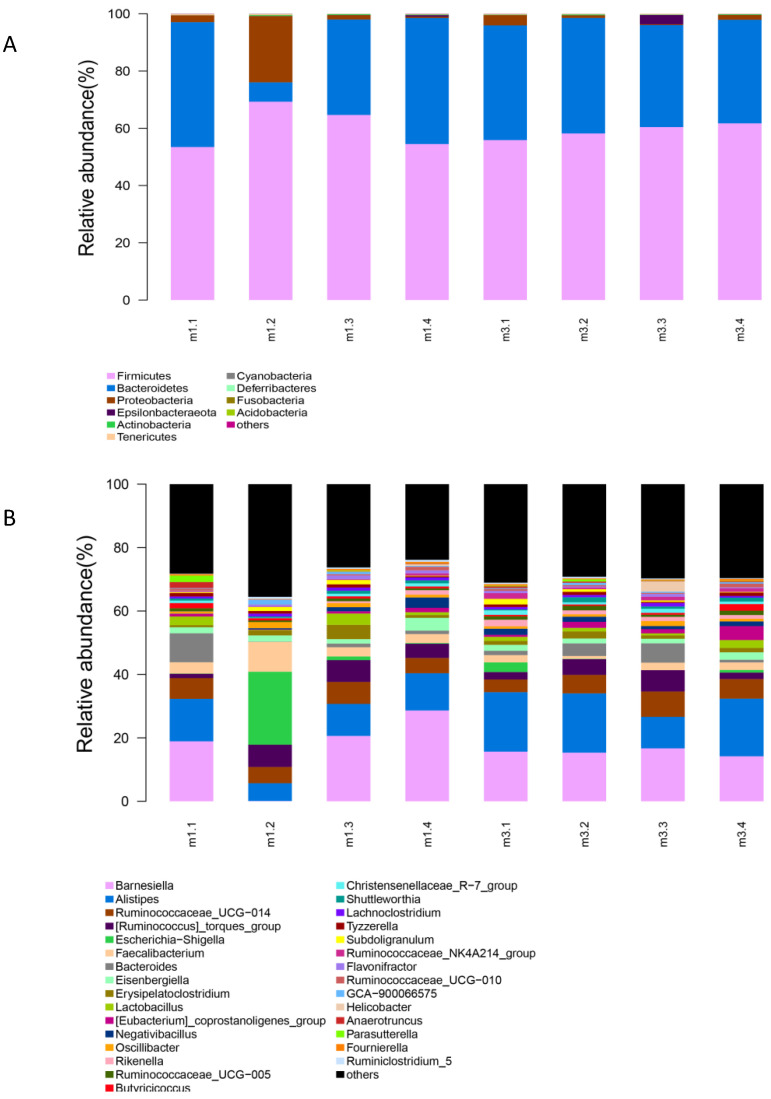
The gut microbiota composition is shown at the phylum level (**A**) and genus level (**B**). m1, control. m3, *E. faecium*.

**Figure 3 animals-10-01232-f003:**
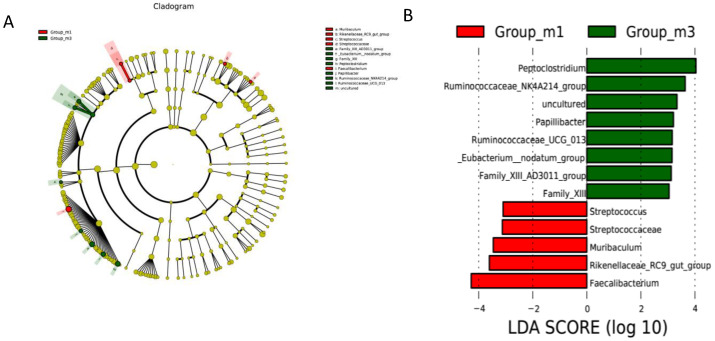
The taxonomic cladogram (**A**) and the LDA (linear discriminant analysis) score (**B**) obtained from linear discriminant analysis effect size (LEfSe) analysis of the gut microbiota in different groups. m1, control. m3, *E. faecium* diet.

**Table 1 animals-10-01232-t001:** Ingredient and nutrient composition of basal broiler diets.

Ingredient	Starter (1–21 Days) (g/kg)	Grower (22–42 Days) (g/kg)
Corn	593.1	604.2
Soybean meal	298.8	288.7
Cotton seed meal	50.0	30.0
Soybean oil	15.1	39.8
L-Lysine	1.5	0.9
*DL*-Methionine	1.4	1.6
Limestone	12.7	10.2
CaHPO_4_	19.4	16.6
NaCl	3.0	3.0
Choline chloride	2.0	2.0
Premix ^1)^	1.3	1.3
Zeolite powder	1.7	1.7
Total	1000	1000
Calculated nutrient level		
Metabolic energy (MJ/kg)	12.35	13.02
Crude protein	211.8	198.4
Calcium	10.1	8.5
Available phosphorus	4.5	4.0
Total phosphorus	6.9	6.3
Lysine	11.4	10.5
Methionine	4.9	4.8
Methionine+Cysteine	8.3	8.1
Threonine	7.7	2.2

^1)^ The premix provided the following per kg diet: vitamin A 10,000 IU, vitamin D_3_ 2000 IU, vitamin E 10 IU, vitamin K_3_ 2.5 mg, vitamin B_1_ 1 mg, vitamin B_2_ 6 mg, vitamin B_3_ 10 mg, vitamin B_5_ 40 mg, vitamin B_6_ 3 mg, vitamin B_11_ 0.3 mg, vitamin B_12_ 0.01 mg, biotin 0.12 mg, Cu (as copper sulphate) 8 mg, Fe (as ferrous sulphate) 80 mg, Mn (as manganese sulphate) 60 mg, Zn (as zinc sulphate) 40 mg, Se (as sodium selenite) 0.15 mg and I (as potassium iodide) 0.35 mg.

**Table 2 animals-10-01232-t002:** Primer sequences of chicken *NaP-IIb*; *PiT-1, 2*; and *β-actin.*

Gene	Primer Sequence (5’-3’)	Accession Number
*NaP-IIb*	F: CTGGATGCACTCCCTAGAGC R: TTATCTTTGGCACCCTCCTG	NM_204474.1
*PiT-1*	F: GCTCGTGGCTTCGTTCTTG R: GACCATTTGACGCCTTTCT	XM_015297502.1
*PiT-2*	F: GCAGCAGATACATCAACTC R: ATTTCCACTCCACCCTC	NM_001305398·1
*β-actin*	F: GAGAAATTGTGCGTGACATCA R: CCTGAACCTCTCATTGCCA	NM_205518.1

**Table 3 animals-10-01232-t003:** Dietary *E. faecium* supplementation and broiler growth performance.

Treatment	BW(g)	ADG(g/d)	ADFI(g/d)	F/G
	Starter (1–21 days)			
Control	771.11 ± 20.45	38.56 ± 1.02	51.04 ± 0.52 ^a^	1.374 ± 0.06
*E. faecium*	792.75 ± 18.28	39.64 ± 0.91	44.75 ± 1.49 ^b^	1.251 ± 0.02
*p* Value	0.460	0.453	0.019	0.102
	Grower (22–42 days)			
Control	2207 ± 38.89	73.73 ± 2.79	152.47 ± 3.69	2.116 ± 0.03
*E. faecium*	2262 ± 25.48	76.36 ± 1.36	151.95 ± 5.07	2.071 ± 0.10
*p* Value	0.267	0.422	0.936	0.670

^a,b^ Mean values with dissimilar superscript letters within the same list are significantly different (*p* < 0.05). BW, body weight; ADG, average daily gain; ADFI, average daily feed intake; F/G, feed/gain.

**Table 4 animals-10-01232-t004:** Dietary *E. faecium* supplementation and excreta ash and P content, serum P concentration and alkaline phosphatase (ALP) of broilers.

Treatment	Ash (g/kg Excreta)	P (g/kg Excreta)	P (mmol/L)	ALP (U/L)
	Starter (1–21 days)			
Control	14.86 ± 0.36	1.24 ± 0.02	1.50 ± 0.06	3291 ± 178 ^b^
*E. faecium*	15.11 ± 0.32	1.31 ± 0.05	1.64 ± 0.04	4099 ± 123 ^a^
*p* Value	0.611	0.151	0.073	0.010
	Grower (22–42 days)			
Control	15.58 ± 0.17	1.38 ± 0.03 ^a^	1.47 ± 0.05	1843 ± 176 ^b^
*E. faecium*	15.68 ± 0.21	1.28 ± 0.01 ^b^	1.52 ± 0.06	2787 ± 166 ^a^
*p* Value	0.172	0.050	0.513	0.005

^a,b^ Mean values with dissimilar superscript letters within the same list are significantly different (*p* < 0.05).

**Table 5 animals-10-01232-t005:** Dietary *E. faecium* supplementation on P and ash content of bone, ash of bone and tibia strength of broilers.

Treatment	P of Bone, %	Ash of Bone, %	Tibia Strength, g
	Starter (1–21 days)		
Control	8.54 ± 0.55	53.51 ± 0.59	12868 ± 798
*E. faecium*	8.80 ± 0.16	54.25 ± 0.83	13408 ± 2440
*p* Value	0.847	0.491	0.840
	Grower (22–42 days)		
Control	8.34 ± 0.34 ^b^	52.16 ± 0.27 ^b^	23632 ± 1253
*E. faecium*	10.99 ± 0.40 ^a^	56.32 ± 1.84 ^a^	22896 ± 1712
*p* Value	0.002	0.035	0.736

^a,b^ Mean values with unlike superscript letters within the same list are significantly different (*p* < 0.05).

**Table 6 animals-10-01232-t006:** Dietary *E. faecium* supplementation and *NaP-IIb* and *PiT-1, 2* mRNA expression in small intestinal segments of broilers.

Treatment	*NaP-* *IIb*	*PiT-1*	*PiT-2*
	Duodenum		
Control	1.000 ± 0.20 ^b^	5.861 ± 4.87	3.969 ± 2.97
*E. faecium*	2.335 ± 0.24 ^a^	6.106 ± 6.58	4.547 ± 0.73
*p* Value	0.013	0.255	0.878
	Jejunum		
Control	3.930 ± 1.13 ^b^	3.703 ± 3.08	1.791 ± 1.34
*E. faecium*	11.291 ± 1.16 ^a^	10.176 ± 4.16	3.705 ± 1.67
*p* Value	0.007	0.279	0.406
	Ileum		
Control	1.117 ± 0.13 ^b^	0.893 ± 0.08 ^b^	1.111 ± 0.05 ^a^
*E. faecium*	4.265 ± 0.59 ^a^	8.87 ± 2.08 ^a^	3.523 ± 0.54 ^b^
*p* Value	0.029	0.019	0.046

^a,b^ Mean values with dissimilar superscript letters within the same list are significantly different (*p* < 0.05).

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
