# Peer review of "Enterococcus faecium Modulates the Gut Microbiota of Broilers and Enhances Phosphorus Absorption and Utilization"

_animals, 2020, doi:10.3390/ani10071232_

Round 1

Reviewer 1 Report

It's a well written manuscript of interest to researchers in poultry nutrition and to some extent microbiologists.

I have included my comments in the attached pdf 

Please respond and revise the manuscript accordingly

Author Response

Dear Reviewer:

    Thank you for taking the time to comment on our manuscript. Your work has been very much appreciated. Thanks for the opportunity to resubmit a revised manuscript. We believe that the revision according to your and other 2 reviewers' comments improved the manuscript.

Comments: It's a well-written manuscript of interest to researchers in poultry nutrition and to some extent microbiologists. I have included my comments in the attached pdf. Please respond and revise the manuscript accordingly.

Response: The new manuscript will be resubmitted in a revised format, in which all your suggestions have been responded to in red color.  

Reviewer 2 Report

  1. in this kind of experiments, you mix gender, and you can consider the age or metabolic stages.

Why did you decide only to use one gender in your study. Then it does not apply to all genders.

2. line 94. reference?line 210 consider revesion. not clearAs it has been....

the addition of a  probiotic....

Author Response

Dear Reviewer:

    Thank you for taking the time to comment on our manuscript. Your work has been very much appreciated. Thanks for the opportunity to resubmit a revised manuscript. We believe that the revision according to your and other 2 reviewers' comments improved the manuscript.

Comment 1: In this kind of experiment, you mix gender, and you can consider the age or metabolic stages. Why did you decide only to use one gender in your study? Then it does not apply to all genders.

Response 1: The reason I use the male broiler in the study was to get consistent results, which were easy to conclude. The conclusion revealed in the current study works for all genders of broilers.

Comment 2: line 94. reference? line 210 consider revision. not clear As it has been....the addition of a  probiotic....

Response 2: I have added the reference in line 94. You can find in the new manuscript in line 100.  I have revised line 210, making it more clear. You can find it in line 216 in the new manuscript.

Reviewer 3 Report

The manuscript evaluates the effects of Enterococcus faecium on the gut microbiota and phosphorus absorption and utilization of broilers. The manuscript is overall well-written, well-presented and contains novel and interesting information for the scientific community. However, I have some concerns:

1) Why did the authors rear the broilers in cages rather than pens? It is always preferable in terms of animal welfare. Same consideration for the lighting programme (23L:1D): according to the welfare regulations, a minimum of 6 hours of darkness is recommended. Please, explain.

2) Please, remove from the "Results" section every sentence in which you discuss the obtained findings (i.e., lines 281-282).

3) The discussion about the gut microbiota may be significantly extended. For example, the authors may compare their descriptive findings in terms of most abundant phyla and genera with the already published literature, and they also need to describe in details the other OTUs differences they identified in their study (i.e., the decrease in Faecalibacterium and Escherichia-Shigella and the linkage between E. faecium and Peptoclostridium and Papillibacter, which are not mentioned in the discussion).

Author Response

Dear Reviewer:

Thank you for taking the time to comment on our manuscript. Your work has been very much appreciated. Thanks for the opportunity to resubmit a revised manuscript. We believe that the revision according to your and other 2 reviewers' comments improved the manuscript.

Comment 1: Why did the authors rear the broilers in cages rather than pens? It is always preferable in terms of animal welfare. The same consideration for the lighting program (23L:1D): according to the welfare regulations, a minimum of 6 hours of darkness is recommended. Please, explain.

Response 1: The reason for using cages not pens is because it is desirable in experiments of this nature to avoid coprophagia, which can only be achieved by housing birds in cages. About the lighting program (23L:1D), I made a mistake in the manuscript writing. I have changed the lighting program in the new manuscript to 16L:8D in line 103. Our experimental farm used the lighting program (16L:8D) in this study, but I referred to an old manual when writing the paper.

Comment 2: Please, remove from the "Results" section every sentence in which you discuss the obtained findings (i.e., lines 281-282).

Response 2: I have changed the position of this sentence. You can find in line 152 in the new manuscript.

Comment 3: The discussion about the gut microbiota may be significantly extended. For example, the authors may compare their descriptive findings in terms of most abundant phyla and genera with the already published literature, and they also need to describe in details the other OTUs differences they identified in their study (i.e., the decrease in Faecalibacterium and Escherichia-Shigella and the linkage between E. faecium and Peptoclostridium and Papillibacter, which are not mentioned in the discussion).

Response 3: I have extended the discussion about the gut microbiota. They were written in red color in the new manuscript.